# BAP1 Downregulates NRF2 Target Genes and Exerts Anti-Tumorigenic Effects by Deubiquitinating KEAP1 in Lung Adenocarcinoma

**DOI:** 10.3390/antiox11010114

**Published:** 2022-01-05

**Authors:** Jong-Su Kang, Le Ba Nam, Ok-Kyung Yoo, Kyeong Lee, Young-Ah Suh, Dalyong Kim, Woo Kyung Kim, Chi-Yeon Lim, Haeseung Lee, Young-Sam Keum

**Affiliations:** 1College of Pharmacy and Integrated Research Institute for Drug Development, Dongguk University, 32 Dongguk-ro, Goyang 10326, Gyeonggi-do, Korea; kjspano@nate.com (J.-S.K.); banamdkh@gmail.com (L.B.N.); aamho@naver.com (O.-K.Y.); kaylee@dongguk.edu (K.L.); 2Department of Biomedical Sciences, Asan Medical Center, The University of Ulsan College of Medicine, Seoul 05505, Gyeonggi-do, Korea; ysuh@amc.seoul.kr; 3Department of Internal Medicine, School of Medicine, Dongguk University, Goyang 10326, Gyeonggi-do, Korea; dykim@dumc.or.kr (D.K.); wk2kim@naver.com (W.K.K.); 4Department of Biostatistics, School of Medicine, Dongguk University, Goyang 10326, Gyeonggi-do, Korea; rachun@hanmail.net; 5College of Pharmacy, Pusan National University, Busan 46241, Gyeongsangnam-do, Korea; haeseung@pusan.ac.kr

**Keywords:** NF-E2-related factor 2 (NRF2), KELCH-like ECH-associated protein 1 (KEAP1), BRCA1-associated protein 1 (BAP1), deubiquitinase (DUB), lung adenocarcinoma (LUAD)

## Abstract

KELCH-ECH-associated protein 1 (KEAP1) is an adaptor protein of Cullin 3 (CUL3) E3 ubiquitin ligase that targets a redox sensitive transcription factor, NF-E2-related factor 2 (NRF2). BRCA1-associated protein 1 (BAP1) is a tumor suppressor and deubiquitinase whose mutations increase the risk of several types of familial cancers. In the present study, we have identified that BAP1 deubiquitinates KEAP1 by binding to the BTB domain. Lentiviral transduction of BAP1 decreased the expression of NRF2 target genes, suppressed the migration and invasion, and sensitized cisplatin-induced apoptosis in human lung adenocarcinoma (LUAD) A549 cells. Examination of the lung tissues in Kras^G12D/+^ mice demonstrated that the level of Bap1 and Keap1 mRNAs progressively decreases during lung tumor progression, and it is correlated with NRF2 activation and the inhibition of oxidative stress. Supporting this observation, lentiviral transduction of BAP1 decreased the growth of A549 xenografts in athymic nude mice. Transcriptome analysis of human lung tissues showed that the levels of Bap1 mRNA are significantly higher in normal samples than LUAD samples. Moreover, the expression of Bap1 mRNA is associated with a better survival of LUAD patients. Together, our study demonstrates that KEAP1 deubiquitination by BAP1 is novel tumor suppressive mechanism of LUAD.

## 1. Introduction

NF-E2-related factor 2 (NRF2) is a transcription factor that plays an important role in the detoxification of reactive oxygen species (ROS) by transcriptional activation of phase II cytoprotective enzymes [1]. Under basal conditions, NRF2 is located in the cytosol and constantly targeted for polyubiquitination by KELCH-like ECH-associated protein 1 (KEAP1), an adaptor protein for Cullin 3 (CUL3) E3 ubiquitin ligase [2]. Exposure of oxidative insults halts polyubiquitination of NRF2, which causes NRF2 to translocate into the nucleus and initiates transcriptional activation of phase II cytoprotective enzymes by binding to the antioxidant response element (ARE), a *cis*-acting motif sequence existing in the promoter of NRF2 target genes [3]. While NRF2 activation has been regarded as a target for chemoprevention and treatment of proinflammatory diseases [4], the studies in the last decade have demonstrated that NRF2 is constitutively activated in cancer due to somatic mutations in the KEAP1/NRF2 pathway, conferring chemoresistance and radioresistance [5]. Therefore, the inhibition of NRF2 is currently considered as an attractive strategy for chemotherapy [6].

Comprehensive genomic analyses have identified that somatic *Keap1* mutations in cancer frequently occur together with the alterations in other tumor suppressor genes or oncogenes, such as *Tp53, Cdkn2a, Pten**,* and *Pik3ca* [7]. Structurally, a homodimer of KEAP1 binds to a single NRF2 at the DLG and ETGE motifs existing in the Neh2 domain [8]. High-throughput sequencing studies have demonstrated that, while Nrf2 mutations predominantly occur within the DLG and ETGE motifs, ~60% of Keap1 mutations occur within the KELCH domain that is responsible for the binding of KEAP1 to the Neh2 domain of NRF2 [9]. Therefore, it is assumed that Keap1 or Nrf2 mutations disrupt the interaction between KEAP1 and NRF2, thereby providing a strong antioxidant environment necessary for the survival of cancer cells. On the other hand, there is evidence that, similar to NRF2, cellular level of KEAP1 is regulated by polyubiquitination [10], and a study demonstrated that the tripartite motif-containing 25 (TRIM25) targets KEAP1 for polyubiquitination during ER stress [11].

Protein ubiquitination is one of the most powerful post-translational modifications, and regulates diverse cellular processes in distinct manners [12]. Deubiquitinase (DUB) is the type of enzyme that removes ubiquitin from target proteins, plays fundamental roles in ubiquitin homeostasis and protein stability, and exhibits different mode of activities [13]. Humans possess ~100 DUB enzymes and they are classified into six structurally distinct families [14], which include five families of cysteine proteases: the ubiquitin-specific proteases (USPs), the ovarian tumor proteases (OTUs), the ubiquitin C-terminal hydrolases (UCHs), the Josephin family and the motif interacting with ubiquitin-containing novel DUB family (MINDY), and a family of metalloprotease, Zn^2+^-dependent JAB/MPN/MOV34 metalloprotease (JAMMs). While many DUBs do not act in isolation but as a part of protein complexes and large molecular machines, such as E3 ubiquitin ligases [15], the target specificity and regulatory mechanisms of DUBs at the molecular level are poorly understood. In an attempt to find out potential DUBs of KEAP1, we have identified that BRCA1-associated protein 1 (BAP1) is a deubiquitinase of KEAP1. We also observed that KEAP1 deubiquitination by BAP1 stabilizes KEAP1, suppresses NRF2 target genes, and promotes oxidative stress, thereby interfering with the growth of lung cancer cells. Finally, we observed that the expression of human Bap1 mRNA was higher in normal lung samples than in lung adenocarcinoma (LUAD) samples, and it was positively associated with a better survival of LUAD patients.

## 2. Materials and Methods

### 2.1. Cell Culture, Chemicals, Antibodies, and Plasmids

Dulbecco’s Modified Eagle Medium (DMEM) and fetal bovine serum (FBS) were purchased from GenDEPOT (Austin, TX, USA). Phosphate-buffered saline (PBS) and penicillin/streptomycin (Pen/Strep) were purchased from WELGENE (Daegu, Korea). A549 cells and 293T cells were purchased from ATCC (Manassas, VA, USA). A549 cells and 293T cells were cultured in DMEM containing 10% FBS and 1% Pen/Strep in 37 °C at 5% CO_2_. JetPEI transfection agent was purchased from Polyplus-transfection (New York, NY, USA). Anti-HA magnetic beads, protein A/G magnetic beads, and mouse or rabbit secondary antibody (G-21040, 31460) were purchased from Thermo Fisher Scientific (Waltham, MA, USA). Monoclonal ANTI-FLAG-HRP antibody (A8592) and anti-FLAG magnetic beads were purchased from Sigma-Aldrich (St. Louis, MO, USA). Antibodies against NRF2 (#12721), Cleaved Caspase-3 (#9661), Cleaved PARP (#5625), Histone H3 (#4499), MYC-Tag (#2276) and HA-Tag (#2999) were purchased from Cell Signaling Technology (Beverly, MA, USA). Cycloheximide, mouse IgG (sc-2025) and antibodies against KEAP 1(sc-365626), β-Actin (sc-47778), 8-OHdG (sc-393871), BAP1 (sc-28383), and GAPDH (sc-47724) were purchased from Santa Cruz Biotechnology (Santa Cruz, CA, USA). A total of 29 human DUB cDNAs were amplified from various human cell lines by RT-PCR and subcloned into pcDNA3 expression vectors. Human Nrf2, Keap1, and ubiquitin cDNAs were amplified from various cell lines. Full-length sequence of human cDNAs was confirmed by DNA sequencing. Site-directed mutagenesis and the deletion of individual domains were conducted using overlapping PCR technique. pcDNA3-Myc-Ubiquitin, pcDNA3-HA-NRF2, and pcDNA3-HA-KEAP1 plasmids were cloned using In-Fusion^®^ HD Cloning Plus kit (Takara Korea Biomedical Inc., Seoul, Korea). DNA oligonucleotides were purchased from Macrogen (Seoul, Korea).

### 2.2. Western Blot Analysis

Cells were washed three times with 1× PBS and cell pellets were collected. Cell pellets were resuspended with 1× RIPA lysis buffer (50 mM Tris-HCl at pH = 8.0, 150 mM NaCl, 1% NP-40, 0.5% deoxycholic acid, 0.1% sodium dodecyl sulfate (SDS), 1 mM Na_3_VO_4_, 1 mM dithiothreitol (DTT), 1 mM phenylmethylsulfonyl fluoride (PMSF)) and incubated on ice for 1 h. After cell lysates were collected by centrifugation at 13,000 rpm for 10 min, protein concentration was measured using a BCA protein assay kit (Thermo Fisher Scientific, Waltham, MA, USA). Cell lysates were resolved by SDS-PAGE and transferred to PVDF membranes (Merck-Millipore Korea, Daejeon, Korea). The membranes were incubated in blocking buffer (5% skim milk in 1× PBS-0.1% Tween-20, PBST) for 1 h and hybridized with appropriated primary antibodies in 1× PBS (1:1000) overnight at 4 °C. After washing three times with 1× PBST for 30 min, the membranes were hybridized with horseradish peroxidase (HRP)-conjugated secondary antibodies (1:5000) for 1 h at 4 °C and washed three times with 1× PBST for 30 min. The membranes were visualized by using enhanced chemiluminescence (ECL) detection system.

### 2.3. Immunoprecipitation

For immunoprecipitation of epitope-tagged plasmids, confluent 293T cells were cotransfected with FLAG-tagged and HA-tagged pcDNA3 vectors for 48 h using JetPEI reagent. Then, 293T cells were lyzed with 1 mL NP-40 buffer (50 mM Tris-HCl at pH 7.5, 150 mM NaCl, 0.5% NP-40, 50 mM NaF, protease inhibitors) for 1 h on ice and cell lysates were incubated with anti-FLAG magnetic beads or anti-HA magnetic beads overnight at 4 °C. Magnetic beads were washed with NP-40 buffer three times and denatured in 2× sample buffer. Eluted samples were resolved by SDS-PAGE and analyzed by Western blot.

For immunoprecipitation of endogenous BAP1, confluent 293T cells were lyzed with 1 mL NP-40 buffer (50 mM Tris-HCl at pH7.5, 150 mM NaCl, 0.5% NsdP-40, 50 mM NaF, protease inhibitors) for 1 h on ice, and cell lysates were incubated with mouse IgG or anti-BAP1 antibody overnight at 4 °C. Protein A/G magnetic beads were added to the tubes and incubated for 4 h at room temperature. Protein A/G magnetic beads were washed with NP-40 buffer three times and denatured in 2× protein sample buffer. Eluted samples were resolved by SDS-PAGE and analyzed by Western blot analysis.

### 2.4. Generation of Stable Cells by Lentival Transduction

pLenti-CMV-GFP-puro lentiviral expression vector and lentiviral helper plasmids (pMD2.G and psPAX2) were acquired from Addgene (Cambridge, MA, USA), The pLenti-CMV-FLAG-BAP1-puro plasmid was created by replacing GFP with FLAG-BAP1. Confluent 293T cells (2.5 × 10^6^ cells/100 mm dish) were transfected with 1 μg of pLenti-CMV–FLAG-BAP1-puro plasmid together with 1 μg of lentiviral helper plasmids (psPAX2 and pMD2.G). At 72 h post-transfection, the viral supernatant was collected and transduced into A549 cells. Transduced A549 cells were selected with puromycin (2 μg/mL) for 48 h.

### 2.5. Fractionation of the Nucleus and the Cytosol

After lentiviral transduction of FLAG-BAP1, A549 cells were lysed with 1mL NP-40 buffer (50 mM Tris-HCl at pH 7.5, 150 mM NaCl, 0.5% NP-40, 50 mM NaF, protease inhibitors) for 1 h on ice. Lysates were centrifuged at 15,000 rpm for 10 min and supernatant was collected as a cytosolic fraction. After washing the remnant pellets three times with NP-40 buffer, the pellets were resuspended in NP-40 buffer and heavily sonicated. Lysates were centrifuged at 15,000 rpm for 15 min and the supernatant was collected as the nuclear fraction. Both nuclear and cytosolic fractions were subjected to Western blot analysis.

### 2.6. Firefly Luciferase Assay

A549 and A549-FLAG-BAP1 cells (2.0 × 10^6^ cells/well) were cotransfected with pGL3-ARE-luciferase and pGL4.74 vectors. After 24 h transfection, cells were washed three times with 1× PBS and lyzed with luciferase lysis buffer (0.1 M potassium phosphate buffer at pH 7.8, 1% Triton X-100, 1 mM DTT, 2 mM EDTA) for 1 h. The firefly luciferase activity monitored by GLOMAX Multi-system (Promega, Madison, WI, USA) and normalized by the renilla luciferase activity.

### 2.7. Real-time Reverse Transcription-Polymerase Chain Reaction (RT-PCR)

Total RNA of A549 cells and mouse lung tissues was extracted by Hybrid-R RNA extraction kit (GeneAll, Seoul, Korea). A total of 1 μg of total RNA was subject to cDNA synthesis, using PrimeScript RT-PCR kit (TAKARA Korea, Seoul, Korea). The real-time RT-PCR analysis was performed using SYBR mix (ELPIS Biotech. Daejeon, Korea) on CFX384 real-time system as recommended by manufacturer (BioRad, Hercules, CA, USA). The mRNA level of individual genes was normalized by that of GAPDH. The sequence of RT-PCR primers are listed in Table 1.

### 2.8. Wound Healing Assay

A549 cells (2.0 × 10^5^ cells/well) and A549-FLAG-BAP1 cells (2.0 × 10^5^ cells/well) were seeded in 24-well plates and a straight wound was created by scratching with a yellow pipette tip. Cells were cultured in serum-free medium and allowed to migrate into the wound area and the images of cell movement were obtained using the Eclipse Ti-U inverted microscope (Nikon, Tokyo, Japan). Wounded areas were calculated using NIS Elements F software (Nikon, Tokyo, Japan).

### 2.9. Cell Migration Assay

Migration assays were performed using transwell inserts (Neuro Probe Inc., Gaithersburg, MD, USA). After lower surface of transwell inserts was immersed in fibronectin solution (10 μg/mL) overnight, A549 and A549-FLAG-BAP1 cells (5 × 10^5^ cells/mL) suspended in serum-free medium were added to the upper chamber of each inserts and the lower chambers were filled up with medium containing 3% FBS. After incubation for 6 h, cells that failed to migrate were scraped off from the upper surface of the membrane and those that reached the lower surface were stained by the Diff Quik staining kit (Sysmex, Kobe, Japan). The image of cells in the lower chamber was observed and captured using the Eclipse Ti-U inverted microscope (Nikon, Tokyo, Japan), and the number of migrated cells was manually counted using the computer.

### 2.10. TUNEL Assay

TUNEL assay was conducted using DeadEND^TM^ fluorometric TUNEL system kit (Promega, Madison, WI, USA) as recommended by the manufacturer.

### 2.11. Animal Experiments

Six-week-old Balb/c nude mice were purchased from Daehan Biolink Co. (Eumseong, Korea) and used for xenografts. After a week acclimation, 12 mice were divided into two groups of six mice, and they were subcutaneously injected into the dorsal flank with A549 cells (5 × 10^6^ cells/0.2 mL DMEM media) or A549-FLAG-BAP1 cells (5 × 10^6^ cells/0.2 mL DMEM media). The body weights of mice were measured every three days during the course of study. After 18 days, mice were sacrificed by asphyxiation with CO_2_ and the weight of xenografts was measured.

Kras^G12D/+^ mice that can form lung tumors by spontaneous recombination events [16] were obtained from Dr. Tyler Jacks (Massachusetts Institute of Technology, Cambridge, MA, USA). Kras^G12D/+^ mice were housed in the sterile filter-capped microisolator cages and provided with water and diet *ad libitum*. After Kras^G12D/+^ mice were bred for 1 week, 8 weeks, 24 weeks, and 40 weeks, they were sacrificed, and the lung tissues were obtained. The lung tissues were stored in the deep freezer for Western blot and RT-PCR or in 10% paraformaldehyde for hematoxylin and eosin staining and immunohistochemistry.

### 2.12. Hematoxylin and Eosin (H&E) Staining and Immunohistochemistry (IHC)

Dehydration of the lung tissues in 10% paraformaldehyde was performed by serially immersing the tissues into 75%, 80%, 85%, 90%, 95%, and 100% ethanol, and xylene solution for 1 h at each step. Dehydrated tissues were embedded in the paraffin block. Paraffin-embedded tissues were sectioned at the length of 5 μm, mounted on the slide, and deparaffinized. To conduct the H/E staining, the tissues were stained with Mayer’s hematoxylin solution for 5 min at room temperature and rinsed in tap water until the water became clear. In the bluing step, the tissues were stained with repeated cycles of eosin Y ethanol solution for 70 s, 5 dips in 95% ethanol, and 5 dips in 100% ethanol at room temperature. The tissues were rinsed with distilled water and the images were taken using the Eclipse Ti-U inverted microscope (Nikon, Tokyo, Japan).

To conduct the IHC, the paraffin-embedded tissues were sectioned at the length of 5 μm, mounted on the slide, and deparaffinized. Tissues on the slide were heated with citrate buffer (pH 6.0) in the microwave for 10 min and the slides were blocked by blocking solution (ScyTek Inc., Logan, UT, USA) for 30 min. After washing three times with 1× PBS, the slides were incubated with primary antibodies (1:200) overnight at 4 °C. After washing three times with 1× PBST, the slides were incubated with anti-mouse UltraTEk HRP antibodies (ScyTek Inc., Logan, UT, USA), and development of the slides was performed with DAB kit (GBI Labs, Bothell, WA, USA). The slides were sealed with a mounting medium and the images were captured using the Eclipse Ti-U inverted microscope (Nikon, Tokyo, Japan). The percentage of stained area under the bright field was monitored by the ImageJ software (National Institute of Health, Bethesda, MD, USA).

### 2.13. Statistics

#### 2.13.1. Statistical Comparison of A549 Cells and A549 Tumor Xenografts

Student’s *t*-test was used to statistically analyze the difference in cell line experiments and A549 tumor xenograft experiments. Asterisks indicate statistical a significance with * *p* < 0.05, ** *p* < 0.01, and *** *p* < 0.001.

#### 2.13.2. Comparison of Bap1 mRNA Expression between LUAD and Normal Lung Tissues of Humans

The levels of Bap1 mRNA expression (Transcripts Per kilobase Million, TPM) from RNA-sequencing (RNA-seq) data of 513 LUAD samples and 288 normal lung samples were obtained from the UCSC Xena [17], where the RNA-seq data of the TCGA and the GTEx were reprocessed by using a uniform RNA-seq analysis pipeline (alignment to hg38 genome and quantification using RSEM). Student’s *t*-test was applied to determine the difference in mean expression levels of Bap1 mRNA between tumors and normal samples.

#### 2.13.3. Survival Analysis of LUAD Cancer Patients

Transcriptome and clinical data of LUAD and non-small cell lung cancer (NSCLC) patients were obtained from the TCGA and the Gene Expression Omnibus (GEO) databases, respectively. In TCGA data, RNA-seq profiles and clinical information were downloaded via the R package “TCGA biolinks”, and TPM was taken as the gene expression level. In each cohort, patients were divided into two groups, high and low, based on whether their expression level of *BAP1* was higher or lower than the median level. Survival analysis was performed to test the difference in overall survival (OS) rates between the high and low groups via the R package “survival”. Hazard ratio (HR) and *p*-value (P) were calculated using Cox proportional hazards regression analysis and log-rank test, respectively.

## 3. Results

### 3.1. Identification of BAP1 as a DUB of KEAP1

In order to identify the DUBs of KEAP1, pcDNA3-FLAG-DUB plasmids were transfected in 293T cells and the level of endogenous KEAP1 was measured by Western blot analysis. Among DUBs, we observed that overexpression of FLAG-BRCA1-associated protein 1 (BAP1) or FLAG-Ubiquitin C-terminal hydrolase L3 (UCHL3) induced KEAP1 (Figure 1A). To examine whether BAP1 and UCHL3 directly interacts with KEAP1, we cotransfected pcDNA3-FLAG-BAP1 or pcDNA3-FLAG-UCHL3 with pcDNA3-HA-KEAP1 in 293T cells and immunoprecipitated cell lysates with FLAG-agarose beads followed by Western blot analysis with HA-HRP antibody. We observed that FLAG-BAP1 (Figure 1B, upper panel), but not FLAG-UCHL3 (Figure 1B, lower panel), directly binds to HA-KEAP1. However, FLAG-BAP1 did not bind to HA-NRF2 (Figure 1C). Immunoprecipitation of endogenous BAP1 demonstrates that BAP1 interacts with endogenous KEAP1 (Figure 1D). Overexpression of pcDNA3-FLAG-BAP1 attenuated polyubiquitination of HA-KEAP1, whereas that of FLAG-BAP1-C91A, a catalytic mutant of BAP1, failed to do so (Figure 1E). These results demonstrate that BAP1 is a DUB of KEAP1.

### 3.2. BAP1 Stabilizes KEAP1 by Binding to the BTB Domain

KEAP1 possesses the BTB (amino acids 79–143), the BACK (amino acids 184–286), and the KELCH domains (amino acids 317–611) (Figure 2A, left panel) [18]. In order to identify the domain of KEAP1 that interacts with BAP1, we generated pcDNA3-HA-KEAP1 mutant plasmids lacking individual domains (HA-KEAP1ΔBTB, HA-KEAP1ΔBACK, and HA-KEAP1ΔKELCH). pcDNA3-HA-KEAP1 and pcDNA3-HA-KEAP1 deletion mutants were cotransfected with pcDNA3-FLAG-BAP1 in 293T cells and cell lysates were immunoprecipitated with HA-beads followed by Western blotting analysis with FLAG-HRP antibody. Our result shows that FLAG-BAP1 does not selectively bind to HA-KEAP1ΔBTB (Figure 2A, right panel), suggesting that BAP1 binds to the BTB domain of KEAP1.

BAP1 consists of the CD (Catalytic Domain, amino acids 1–240), the CC1 (Coiled-Coil Domain 1, amino acids 240–265), the CL (Crossover Loop, amino acids 266–529), the CC2 (Coiled-Coil Domain 2, amino acids 530–654), the ULD (UCH37-like Domain, amino acids 654–699), and the CTE (C-terminal Extension, amino acids 699–729) domains (Figure 2B, left panel) [19]. In order to identify the domains of BAP1 that interact with KEAP1, we generated FLAG-BAP1 mutant plasmids lacking individual domains (FLAG-BAP1ΔCD, FLAG-BAP1ΔCC1, FLAG-BAP1ΔCL, FLAG-BAP1ΔCC2, FLAG-BAP1ΔULD, and FLAG-BAP1ΔCTE). pcDNA3-FLAG-BAP1 and pcDNA3-FLAG-BAP1 deletion mutants were cotransfected with pcDNA3-HA-KEAP1 in 293T cells and cell lysates were immunoprecipitated with FLAG-agarose beads followed by Western blotting analysis with HA-HRP antibody. Our result shows that HA-KEAP1 fails to bind to FLAG-BAP1ΔCL (Figure 2B, right panel), suggesting that KEAP1 binds to the CL domain in BAP1.

Because BAP1 bound to the BTB domain (Figure 2A) and promoted deubiquitination of KEAP1 (Figure 1E), we assumed that the BTB domain might be responsible for maintaining the protein stability of KEAP1 by BAP1. To address this issue, we generated 293T-HA-KEAP1 and 293T-HA-KEAP1ΔBTB cells by stable transfection of pcDNA3-puro-HA-KEAP1 and pcDNA3-puro-HA-KEAP1ΔBTB plasmids followed by selection with puromycin. Established 293T-HA-KEAP1 and 293T-HA-KEAP1ΔBTB cells were transfected with pcDNA3-FLAG-BAP1 and exposed to cycloheximide (CHX) to inhibit de novo protein synthesis. Our results show that treatment of CHX decreased the level of HA-KEAP1, and this event was significantly attenuated by overexpression of FLAG-BAP1 in 293T-HA-KEAP1 cells (Figure 2C, left panel). However, FLAG-BAP1 failed to attenuate a decrease in the level of HA-KEAP1ΔBTB induced by treatment of CHX in 293T-HA-KEAP1ΔBTB cells (Figure 2C, right panel), suggesting that the BTB domain is responsible for maintaining the protein stability of KEAP1 by BAP1.

### 3.3. Overexpression of BAP1 Inhibits NRF2, Suppresses Migration and Invasion, and Promotes Cisplatin-Induced Apoptosis of A549 Cells

Aberrant NRF2 activity is frequently observed in human LUAD due to loss of function mutations in Keap1 [20]. Thus, we examined whether BAP1 can affect KEAP1 to modulate NRF2 target gene expression in human LUAD A549 cells. To address this issue, we generated stable A549-FLAG-BAP1 cells by lentiviral transduction and fractionated them into the nucleus and the cytosol. Our results show that transduced FLAG-BAP1 is mostly located in the cytosol (Figure 3A). We also observed that A549-FLAG-BAP1 cells exhibited a lower level of ARE-luciferase activity compared with A549 cells (Figure 3B). Transfection of pcDNA3-FLAG-BAP1 increased the level of KEAP1 and decreased the level of NRF2 in A549 cells (Figure 3C). Real-time RT-PCR analysis demonstrates that lentiviral transduction of FLAG-BAP1 significantly decreased the mRNA level of NRF2 target genes such as heme oxygenase-1 (Hmox1), NADPH:quinone oxidoreductase-1 (Nqo1), glutamate cysteine ligase catalytic subunit (Gclc), and aldo-keto reductase family 1 member B10 (Akr1b10) in A549 cells (Figure 3D). In addition, lentiviral transduction of FLAG-BAP1 suppressed the migration (Figure 3E) and the invasion (Figure 3F) of A549 cells, and potentiated cisplatin-induced apoptosis in A549 cells (Figure 3G,H). Together, these results illustrate that BAP1 lowers the level of NRF2 target genes by deubiquitinating and stabilizing KEAP1, thereby inhibiting the migration and invasion of A549 cells, and sensitizing A549 cells to cisplatin-induced apoptosis.

### 3.4. The Level of Bap1 and Keap1 mRNA Decreases during Progression of Lung Adenocarcinoma in Kras^G12D/+^ Mice and It Is Positively Associated with NRF2 Activation and Suppression of Oxidative Stress

Direct inhibition of KRAS oncoprotein is a difficult task and, therefore, targeting Kras-mutant lung cancer remains a major challenge [21]. It is known that oncogenic KRAS stimulates transformation [22] and lung cancer cell proliferation, at least in part, by induction of NRF2 [23]. However, the molecular mechanisms underlying how oncogenic KRAS activates NRF2 are still elusive. We hypothesized that oncogenic KRAS might activate NRF2, at least in part, by downregulating BAP1 and KEAP1. To examine this hypothesis, we compared the levels of Bap1, Keap1, Nrf2, and Hmox1 mRNAs in the lung tissues of Kras ^G12D/+^ mice with those of wild-type littermates by real-time RT-PCR analysis. We found that the level of Bap1 and Keap1 mRNAs in the lung tissues of Kras ^G12D/+^ mice progressively decreased from 8 weeks to 40 weeks, compared with those of wild-type littermates (Figure 4A). Consistent with this, the relative mRNA level of Hmox1, a target gene of NRF2, progressively increased from 24 weeks to 40 weeks in the lung tissues of Kras ^G12D/+^ mice (Figure 4A). However, the relative level of Nrf2 mRNA in the lung tissues of Kras ^G12D/+^ mice was not different from that of wild-type littermates (Figure 4A), supporting that the level of NRF2 is regulated mostly by CUL3/KEAP1-mediated polyubiquitination [24]. Western blot analysis shows that the levels of BAP1 and KEAP1 were significantly lower in the lung tissues of Kras ^G12D/+^ mice at 40 weeks, when compared with those of wild-type littermates (Figure 4B). Immunohistochemistry results demonstrate that the levels of BAP1, KEAP1, and 8-hydroxydeoxyguanosine (8-OH-dG), an oxidative damage marker, significantly decreased in the lung tumors of Kras^G12D/+^ mice, compared with normal tissues in the surrounding area (Figure 4C). Together, these results demonstrate that KRAS activation downregulates BAP1 and KEAP1, and this event is correlated with NRF2 activation and a decrease in oxidative stress in vivo.

### 3.5. BAP1 Suppresses the Growth of A549 Xenografts, and the Expression of Bap1 Is Higher in Normal Lung Tissues and Correlated with a Better Survival of LUAD Patients

We have demonstrated that BAP1 inhibits the expression of NRF2 target genes (Figure 3D) in A549 cells by promoting KEAP1 deubiquitination (Figure 1E). To examine whether BAP1 can affect the growth of A549 cells in vivo, A549 and A549-FLAG-BAP1 cells were injected into athymic nude mice, and the growth of xenografts was monitored. While athymic nude mice bearing A549 cells and A549-FLAG-BAP1 cells exhibited no difference in the body weight (Figure 5A), we observed that the mice bearing A549-FLAG-BAP1 produced fewer and smaller xenografts at sacrifice (Figure 5B). To examine whether BAP1 might behave as a tumor suppressor of LUAD, we retrieved the RNA-seq data of LUAD and normal lung tissues in humans from The Cancer Genome Atlas (TCGA) and the Genotype-Tissue Expression (GTEx) projects, respectively, and revealed that the levels of Bap1 mRNA expression were significantly higher in normal samples than in LUAD patients (Figure 5C). In addition, the mRNA level of Bap1 in LUAD patients was positively correlated with an increase in the overall survival (Figure 5D, left panel). It is known that cisplatin is a chemotherapeutic agent which is mostly administered to LUAD patients at stages II–V, but not at stage I [25]. Because BAP1 sensitized A549 cells to cisplatin-induced apoptosis (Figure 3G,H), we subdivided LUAD patients into stage I and stages II–IV and investigated the association between Bap1 mRNA expression and the overall survival rate. Our results showed that the level of Bap1 mRNA was positively correlated with a better survival of LUAD patients at stages II–IV, but not at stage I (Figure 5D). This provides a feasible possibility that BAP1 might provide a clinical benefit to LUAD patients at higher tumor stages.

## 4. Discussion

Inactivating mutations or truncation of Bap1 frequently occur in familial cancers such as uveal melanoma, mesothelioma, cutaneous melanoma, and renal cell carcinoma [26,27]. Mice lacking Bap1 also exhibit myeloid transformation [28] and spontaneous malignant mesothelioma [29], demonstrating that BAP1 is a bona fide tumor suppressor. BAP1 was initially identified as a protein that binds to the RING finger domain of BRCA1 [30]. Follow-up studies have demonstrated that BAP1 interacts with chromatin-associated proteins such as host cell factor 1 (HCF1), O-linked *N*-acetylglucosamine transferase (OGT), and additional sex comb-like proteins (ASXL1 and ASXL2) to regulate the gene expression, double-strand DNA repair, and DNA duplication by deubiquitinating histone H2A at Lysine 119 [31]. On the other hand, the localization of BAP1 in the cytosol is tightly controlled by the E3 ligase UBE2O [32], where BAP1 interacts with type 3 inositol-1,4,5-triphosphoate receptor (IP3R3) at the endoplasmic reticulum (ER) and promotes Ca^2+^ signaling and apoptosis in response to oxidative stress [33]. While the level of BAP1 was very minimal in A549 cells (data not shown), we observed that transduced BAP1 was mostly located in the cytosol (Figure 3A), stabilized KEAP1 (Figure 3C), and suppressed the expression of NRF2 target genes (Figure 3D). At present, we are unaware whether UBE2O played a role in the cytoplasmic localization and/or the stability of BAP1 in A549 cells.

It was previously demonstrated that USP15 is a DUB of KEAP1 [34]. However, USP15 failed to induce KEAP1 (Figure 1A) nor bind to KEAP1 in our hands (data not shown). In the present study, we have demonstrated that BAP1 deubiquitinates KEAP1 (Figure 1E) by binding to the BTB domain (Figure 2A) and that KEAP1 lacking the BTB domain fails to respond to the stabilization by BAP1 (Figure 2C). The molecular mechanisms underlying how the BTB domain contributes to the stability of KEAP1 are largely unknown, but it is possible to assume that the BTB domain might possess potential lysine residue(s) susceptible for polyubiquitination of KEAP1 that can be removed by BAP1. Alternatively, it can be speculated that BAP1 might competitively bind to KEAP1 at the BTB domain with unknown protein(s) affecting the stability of KEAP1. However, it is unlikely that TRIM25, an E3 ubiquitin ligase of KEAP1, competes with BAP1 because TRIM25 is reported to bind to KEAP1 at the KELCH domain, not the BTB domain [11]. On the other hand, we observed that KEAP1 binds to the CL domain in BAP1 (Figure 2B). While the crystal structure of full-length BAP1 is unavailable, De et al. have provided the crystal structure of a BAP1 orthologue from *Drosophila melanogaster*, Calypso, bound to its chromatin activator ASX, additional sex combs [19]. However, the structure of the CL domain in Calypso was removed from this study because the CL domain exhibited a large conformational flexibility and, therefore, interfered with the crystallization of Calypso.

It is known that RAS activation accounts for 30% human LUAD, in which Keap1 is frequently mutated or deleted [35]. This suggests a possibility that the inhibition of KEAP1 might be a prerequisite event necessary for promoting LUAD caused by RAS activation. Supporting this speculation, we observed that the levels of Bap1 and Keap1 mRNA were progressively decreased in the lung tissues of Kras^G12D/+^ mice, compared with those of wild-type littermates during tumor progression (Figure 4A). At present, the mechanisms by which KRAS activation downregulates BAP1 and KEAP1 in vivo are unclear. Because RAS activation induces the activation of various intracellular signaling kinases to promote the growth of tumors [36], it would be interesting to find out whether and, if so, how the activation of these kinases caused by RAS contributes to the inhibition of Bap1 and Keap1 transcription. Examination of the TCGA database also showed that the level of Bap1 mRNA was lower in the lung tissues of human LUAD compared with those of normal people (Figure 5C) and it was associated with an increase in the survival of LUAD patients at tumor stages II–IV (Figure 5D). These results illustrate that genetic or pharmacological BAP1 activation could be an effective strategy to treat LUAD patients at advanced tumor stages. Collectively, our study demonstrates that BAP1 is a DUB of KEAP1. While BAP1 has been regarded as a tumor suppressor possessing diverse molecular mechanisms [37], we reveal a possibility that BAP1 might serve as a redox regulator to inhibit NRF2 target genes and suppress the growth of lung cancer cells by deubiquitinating KEAP1. This finding provides a feasible possibility that targeting NRF2 might be helpful in treating familial cancers arising from Bap1 deficiency or mutations.

## 5. Conclusions

We have demonstrated that BAP1 directly binds to, deubiquitinates, and stabilizes KEAP1, exerting tumor suppressive effects in vitro. KRAS activation leads to a decrease in the levels of BAP1 and KEAP1 in the lung, resulting in NRF2 activation. Bioinformatics analyses indicate that the level of BAP1 is higher in normal human lung tissues and is positively associated with a better survival LUAD patients. Our results demonstrate that BAP1 is a tumor suppressor, which suppresses NRF2 in LUAD by deubiquitinating KEAP1.

## Figures and Tables

**Figure 1 antioxidants-11-00114-f001:**
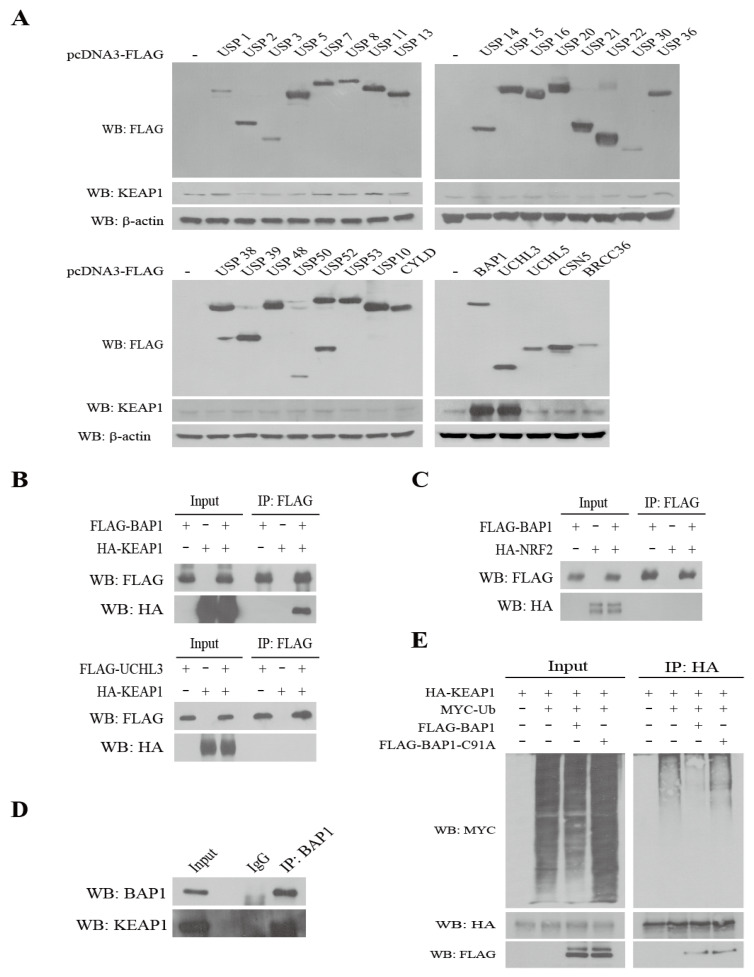
BAP1 directly binds to and deubiquitinates KEAP1. (**A**) Overexpression of BAP1 and UCHL3 induces KEAP1 in 293T cells. One μg of pcDNA3-FLAG-DUBs was transfected to 293T cells (2.0 × 10^6^ cells/100 mm dish) for 24 h and Western blot analysis conducted against KEAP1. (**B**) FLAG-BAP1 directly binds to HA-KEAP1. One μg of pcDNA3-FLAG-BAP1 (**upper panel**) or 1 μg of pcDNA3-FLAG-UCHL3 (**lower panel**) were cotransfected with 1 μg of pcDNA3-HA-KEAP1 for 24 h in 293T cells (2.0 × 10^6^ cells/100 mm dish). The 293T cell lysates were immunoprecipitated with FLAG magnetic beads, and Western blot analysis was performed with FLAG-HRP and HA-HRP antibodies. (**C**) FLAG-BAP1 does not bind to HA-NRF2. One μg of pcDNA3-FLAG-BAP1 was cotransfected with 1 μg of pcDNA3-HA-NRF2 for 24 h in 293T cells (2.0 × 10^6^ cells/100 mm dish). Cell lysates were immunoprecipitated with FLAG magnetic beads, and Western blot was performed with FLAG-HRP and HA-HRP antibodies (1:2000). (**D**) Endogenous BAP1 binds to endogenous KEAP1. 293T cells (4 × 10^6^ cells/100 mm dish) were cultured and lyzed with RIPA buffer on ice. Cell lysates (1 mg) were immunoprecipitated with mouse IgG (1 μg) and BAP1 (1 μg) antibodies, and Western blot analysis was conducted against KEAP1. (**E**) BAP1 deubiquitinates KEAP1. One μg of pcDNA3-HA-KEAP1 and 1 μg of pcDNA3-Myc-Ubiquitin were cotransfected with 1 μg of pcDNA3-FLAG-BAP1 or 1 μg of pcDNA3-FLAG-BAP1-C91A (a catalytic mutant of BAP1) for 24 h in 293T cells (2.0 × 10^6^ cells/100 mm dish). Immunoprecipitation was conducted with HA magnetic beads and Western blot analysis was performed against Myc, HA-HRP, and FLAG-HRP antibodies, respectively.

**Figure 2 antioxidants-11-00114-f002:**
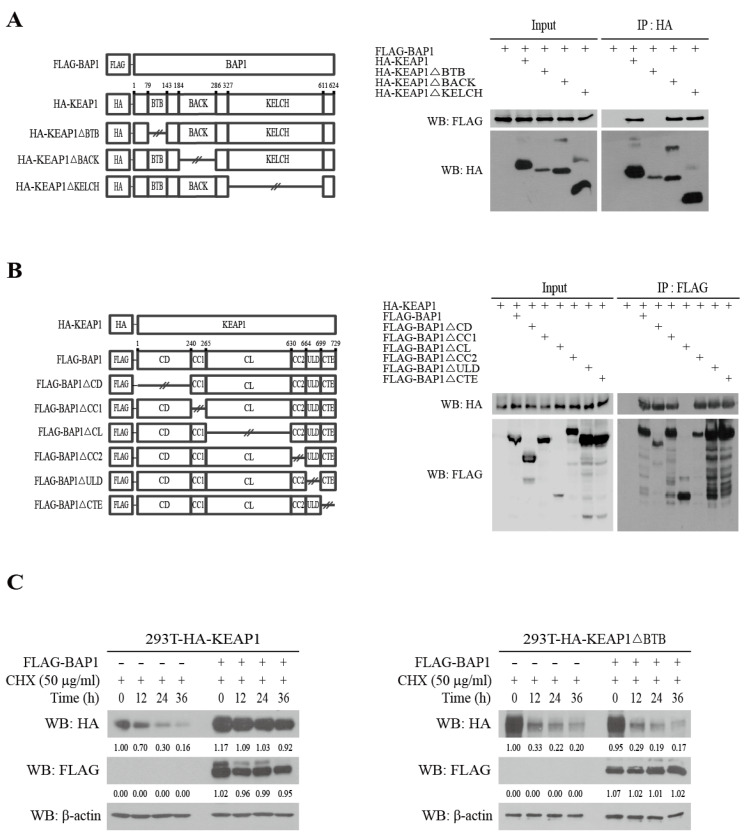
BAP1 stabilizes KEAP1 by binding to the BTB domain. (**A**) BAP1 binds to the BTB domain of KEAP1. Individual KEAP1 domains in pcDNA3-HA-KEAP1 plasmids were deleted by overlapping PCR (**left panel**). One μg of pcDNA3-FLAG-BAP1 was cotransfected with 1 μg of pcDNA3-HA-KEAP1 or 1 μg of pcDNA3-HA-KEAP1 truncated mutants for 24 h in 293T cells (2.0 × 10^6^ cells/100 mm dish). Cell lysates (1 mg) were immunoprecipitated with HA magnetic beads and Western blot was performed with FLAG-HRP and HA-HRP antibodies (**right panel**). (**B**) KEAP1 binds to the CL domain of BAP1. Individual BAP1 domains in pcDNA3-FLAG-BAP1 plasmids were deleted by overlapping PCR (**left panel**). One μg of pcDNA3-HA-KEAP1 was cotransfected with 1 μg of pcDNA3-FLAG-BAP1 or 1 μg of pcDNA3-FLAG-BAP1 truncated mutants in 293T cells (2.0 × 10^6^ cells/100 mm dish). Cell lysates were immunoprecipitated with FLAG magnetic beads, and Western blot analysis was conducted with HA-HRP and FLAG-HRP antibodies (**right panel**). (**C**) The BTB domain is necessary for the stabilization of KEAP1 by BAP1. The 293T-HA-KEAP1 cells and 293T-HA-KEAP1ΔBTB cells (2.0 × 10^6^ cells/100 mm dish) were established by transfection of pcDNA3-puro-HA-KEAP1 and pcDNA3-puro-HA-KEAP1ΔBTB plasmids followed by selection with puromycin (2 μg/mL) for 48 h. After transfection of 1 μg of pcDNA3-FLAG-BAP1, 293T-HA-KEAP1 cells (**left panel**) and 293T-HA-KEAP1ΔBTB cells (**right panel**) were exposed to CHX for various times and Western blot analysis was conducted against HA-HRP and FLAG-HRP antibodies.

**Figure 3 antioxidants-11-00114-f003:**
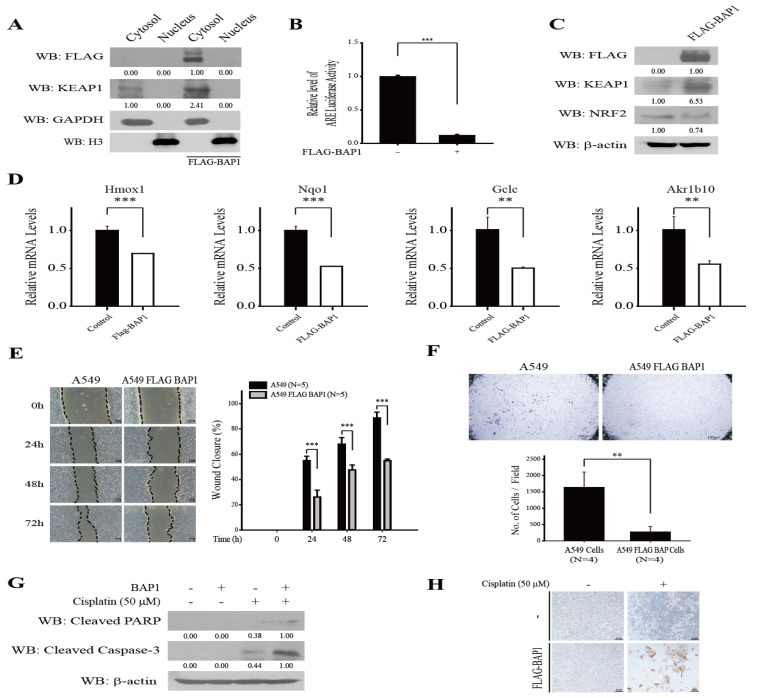
Overexpression of BAP1 suppresses NRF2 targets genes, inhibits the migration and invasion, and sensitizes cisplatin-induced apoptosis in A549 cells. (**A**) Lentivirally-transduced BAP1 is located in the cytosol of A549 cells. A549-FLAG-BAP1 cells were established by lentiviral transduction with pLenti-puro-FLAG-BAP1 followed by selection with puromycin. A549-FLAG-BAP1 cells (2.5 × 10^6^ cells/100 mm dish) were fractionated into the nucleus and the cytosol, and Western blot analysis was conducted against FLAG-HRP, KEAP1, GAPDH, and histone H3. GAPDH and Histone H3 were used as indicative markers for the cytosol and the nucleus, respectively. (**B**) FLAG-BAP1 suppresses ARE-luciferase activity. A549 cells (5 × 10^5^ cells/24 well dish) and A549-FLAG-BAP1 cells (5 × 10^5^ cells/24-well dish) were cotransfected with 0.1 μg of pGL3-ARE-luciferase and 0.1 μg of pGL4.74 vectors, and the firefly luciferase activity was measured. The firefly luciferase activity was normalized by the renilla luciferase activity. (**C**) Overexpression of BAP1 increases KEAP1 and decreases NRF2 in A549 cells. A549 cells (2.5 × 10^6^ cells/100 mm dish) were transfected with 1 μg of pcDNA3-FLAG-BAP1 and Western blot analysis was conducted against FLAG, KEAP1, and NRF2. (**D**) The relative mRNA level of NRF2 target genes (Hmox1, Nqo1, Gclc, Akr1b10) was compared between A549 cells (5 × 10^6^ cells/100 mm dish) and A549-FLAG-BAP1 cells (5 × 10^6^ cells/100 mm dish) by real-time RT-PCR analysis. The primer sequence of individual genes is provided in Table 1. (**E**) BAP1 suppresses the migration of A549 cells. The wound closure of A549 cells and A549-FLAG-BAP1 cells was monitored for 24 h, 48 h, and 72 h (**left panel**), and the percentage of wound closure was plotted (**right panel**). (**F**) BAP1 suppresses the invasion of A549 cells. The number of migrated cells to the lower chamber was manually counted under the microscope after 6 h (**upper panel**) and plotted (**lower panel**). (**G**) Western blot analysis demonstrates that BAP1 sensitizes A549 cells to cisplatin-induced apoptosis. A549 cells were transfected with 1 μg of pcDNA3-FLAG-BAP1 and exposed to cisplatin for 24 h. Western blot analysis was conducted with Cleaved PARP and Cleaved Caspase-3 antibodies. (**H**) TUNEL assay indicates that BAP1 sensitizes A549 cells to cisplatin-induced apoptosis. A549 cells were transfected with 1 μg of pcDNA3-FLAG-BAP1 and exposed to cisplatin for 24 h. TUNEL assay was conducted to visualize the presence of DNA fragmentation in A549 cells. ** *p* < 0.01, *** *p* < 0.001.

**Figure 4 antioxidants-11-00114-f004:**
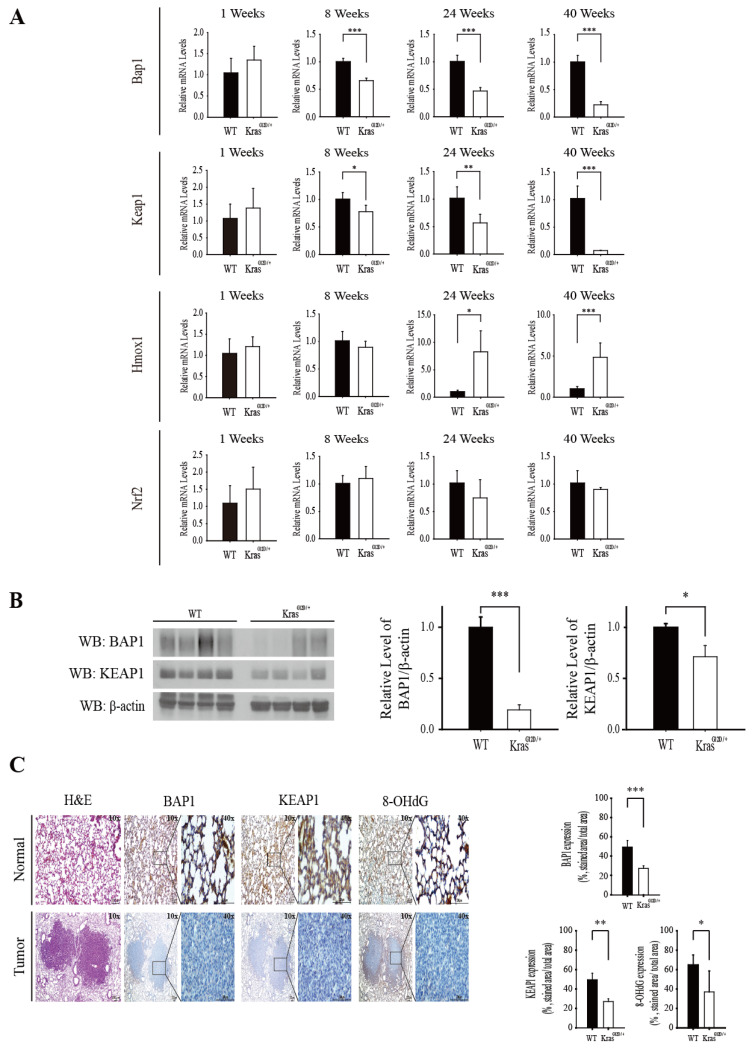
KEAP1 and BAP1 progressively decrease during lung tumor formation in Kras ^G12D/+^ mice, contributing to NRF2 activation and the inhibition of oxidative stress. (**A**) The level of Bap1 and Keap1 mRNAs progressively decreases and that of Hmox1, a target gene of NRF2, progressively increases during lung tumor formation in Kras ^G12D/+^ mouse. Wild-type mice (*n* = 16) and Kras ^G12D/+^ mice (*n* = 16) were sacrificed after 1 week (*n* = 4 per group), 8 weeks (*n* = 4 per group), 24 weeks (*n* = 4 per group), and 40 weeks (*n* = 4 per group). The lung tissues were excised, and real-time RT-PCR was conducted. WT indicates wild-type. (**B**) The level of BAP1 and KEAP1 proteins are lower in the lung tissues of Kras ^G12D/+^ mouse compared with those of wild-type littermates. The lung tissues of 40-week old wild-type and Kras ^G12D/+^ mice were subjected to Western blot analysis against BAP1 and KEAP1 antibodies (**left panel**), and the relative level of BAP1 and KEAP1 in wild-type mice and Kras ^G12D/+^ mice is plotted (**right panel**). WT indicates wild-type. (**C**) The levels of BAP1, KEAP1, and 8-OHdG are lower in the lung tumors of Kras ^G12D/+^ mice, compared with normal lung tissues of wild-type (WT) littermates. The lung tissues of 40-week-old normal mice and Kras ^G12D/+^ mice were subjected to H&E staining and immunohistochemistry against BAP1, KEAP1, and 8-OHdG. Representative figures are provided (**left panel**). The relative staining intensity of BAP1, KEAP1, and 8-OHdG in the normal lung tissue (*n* = 4) and lung tumor (*n* = 4) areas is plotted (**right panel**). * *p* < 0.05, ** *p* < 0.01, and *** *p* < 0.001.

**Figure 5 antioxidants-11-00114-f005:**
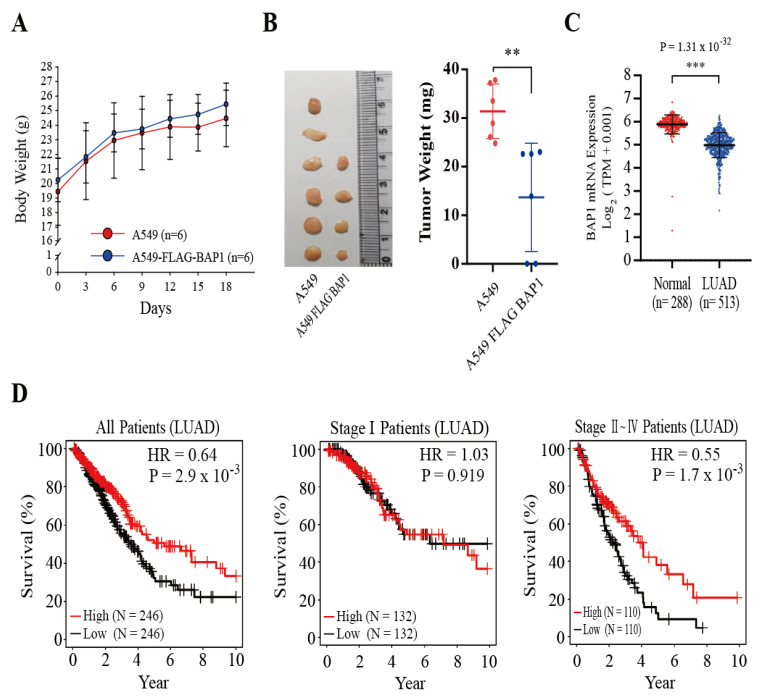
BAP1 suppresses the growth of A549 tumor xenografts and behaves as a tumor suppressor of LUAD. (**A**) The body weight of mice bearing A549 xenografts and A549-FLAG-BAP1 xenografts is indifferent. Athymic nude mice were injected with A549 cells (*n* = 6) and A549-FLAG-BAP1 cells (*n* = 6), and the body weights of mice were measured during the course of study. (**B**) Athymic mice bearing A549-FLAG-BAP1 xenografts produced fewer and smaller xenografts than mice bearing A549 xenografts. At sacrifice, the number (left panel) and the weight (right panel) of A549 and A549-FLAG-BAP1 xenografts were measured. (**C**) The expression of *BAP1* mRNA levels is higher in normal lung samples than in LUAD samples. The mRNA levels (TPM) of *BAP1* in normal and LUAD samples were obtained from the GTEx and the TCGA studies, respectively, in UCSC Xena (https://xena.ucsc.edu/, accessed on 25 March 2021). (**D**) The Kaplan–Meier curves show overall survival rate of LUAD patients at all stages (**left panel**), at stage I (**middle panel**), and at stages II–IV (**right panel**) stratified by BAP1-high and -low groups. ** *p* < 0.01, *** *p* < 0.001.

**Table 1 antioxidants-11-00114-t001:** List of real-time PCR primers.

	Accession No.	Gene	Primer Sequence
Mouse	NM_027088	BAP1	Forward: 5′-CAGTGAGCCCTTGAGTGGAG-3′ Reverse: 5′-GGTCCTTCGCTGGTCATCAA-3′
	NM_016679	KEAP1	Forward: 5′-CTCAACCGCTTGCTGTATGC-3′ Reverse: 5′-TTCAACTGGTCCTGCCCATC-3′
	NM_010442	HMOX1	Forward: 5′-AGCCCCACCAAGTTCAAACA-3′ Reverse: 5′-TCTCTGCAGGGGCAGTATCT-3′
	NM_010902	NRF2	Forward: 5′-CGCTGGAAAAAGAAGTGGGC-3′ Reverse: 5′-GTGACAGGTCACAGCCTTCA-3′
	NM_01289726	GAPDH	Forward: 5′-GGAGAGTGTTTCCTCGTCCC-3′ Reverse: 5′-ACTGTGCCGTTGAATTTGCC-3′
Human	NM_002133	HMOX1	Forward: 5′-GTGCCACCAAGTTCAAGCAG-3′ Reverse: 5′-CAGCTCCTGCAACTCCTCAA-3′
	NM_000903	NQO1	Forward: 5′-GGTTTGGAGTCCCTGCCATT-3′ Reverse: 5′-GCCTTCTTACTCCGGAAGGG-3′
	NM_001197115	GCLC	Forward: 5′-GGAGGAACAATGTCCGAGTT-3′ Reverse: 5′-AGCGAGGGTGCTTGTTTATT-3′
	NM_020299	AKR1B10	Forward: 5′-GACTGTGCCTATGTCTATCA-3′ Reverse: 5′-AAGATAGACGTCCAGATAGC-3′
	NM_002046	GAPDH	Forward: 5′-CATCAATGGAAATCCCATCA-3′ Reverse: 5′-GGCAGAGATGATGACCCTTT-3′

## Data Availability

The data presented in this study are available in this manuscript.

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
