# Peer review of "BAP1 Downregulates NRF2 Target Genes and Exerts Anti-Tumorigenic Effects by Deubiquitinating KEAP1 in Lung Adenocarcinoma"

_antioxidants, 2022, doi:10.3390/antiox11010114_

Round 1

Reviewer 1 Report

This is a nice study done by the authors where a role of BRCA-associated protein 1 (BAP1) was demonstrated in lung adenocarcinomas. The study involved the identification of molecular mechanisms (deubiquitinating KEAP1), feasibility of the BAP1 targeting in-vivo xenograft models, and clinical correlation with TCGA data-set. The manuscript is worth to publish in this journal. However, I would suggest the authors to clarify: why use of BAP1 is better than targeting of other downstream pathways, such as KEAP1 or NRF2 genes. It would not be feasible to use BAP1 itself as a drug candidate unless we can use a system to stimulate the cancer cells to overexpress BAP1.

Author Response

Reviewer 1

This is a nice study done by the authors where a role of BRCA-associated protein 1 (BAP1) was demonstrated in lung adenocarcinomas. The study involved the identification of molecular mechanisms (deubiquitinating KEAP1), feasibility of the BAP1 targeting in-vivo xenograft models, and clinical correlation with TCGA data-set. The manuscript is worth to publish in this journal. However, I would suggest the authors to clarify: why use of BAP1 is better than targeting of other downstream pathways, such as KEAP1 or NRF2 genes. It would not be feasible to use BAP1 itself as a drug candidate unless we can use a system to stimulate the cancer cells to overexpress BAP1.

Response

I would like to appreciate this reviewer for critical and positive comments on our manuscript. I agree with the opinions of this reviewer that targeting downstream pathways such as KEAP1 or NRF2 would be more feasible to treat cancer cells exhibiting NRF2 activation than BAP1. If KEAP1 is targeted, however, it would stimulate NRF2 and it is not appropriate for inhibiting NRF2 in cancer cells. Likewise, targeting the interface between KEAP1 and NRF2 would show analogous effects and, therefore, it is inappropriate. In addition, targeting NRF2 is an impossible task at present because crystal structure of NRF2 is not available. That is why I proposed BAP1 as an alternative target.

Reviewer 2 Report

In the manuscript “BAP1 downregulates NRF2 target genes and exerts anti-tumorigenic effects by deubiquitinating KEAP1 in lung adenocarcinoma.” by Kang J. et al, the authors identified BAP1 a novel regulator of Keap1/NRF2 pathway, which can have cancer implications in lung adenocarcinoma.

The Keap1/NRF2 pathway has been explored in several types of human cancer in the recent years. The authors found a novel regulatory protein for this pathway.

In general, the manuscript is well written, experiments are well designed to answer all the scientific questions raised in the study and results are well interpreted. It is a great work.

I have minor concerns that need to be addressed by the authors to avoid some confusions, add extra information to other researchers in the field can replicate the experiments if they need and to make data clear to the readers. Please consider my comments as positive suggestions to improve the quality of the manuscript (scientifically, but also to help who will read the manuscript when published).

A) In the materials and methods section, the authors should include the manufacturer’s references for the purchased reagents, whenever possible. Sometimes there are similar products with slight formulation differences; other example is the fact that there are several antibodies for the same protein from the same company; and so on. So please, whenever possible, please add the product reference.

B) The authors describe the antibodies used, but they should mention the dilution made. Please add this extra information!

C) In the WB method description, I am not sure about these points.

C1) Usually the incubation with secondary antibody is made at room temperature. Do the authors incubate for 1h at 4ºC as described?

C2) Do the authors dilute primary antibodies only in 1x PBS? Or do they dilute in 5% non-fat milk (in 1x PBS)?

C3) Which ECL system do authors use? Please add the company and reference of the product.

C4) Please add antibody dilutions (primary and secondary antibodies).

D) There is a missing information in the material and methods section for some plasmids used. For instance, there is no mention to the HA-NRF2 construct (figure 1C). Also for some others.

E) Please add additional information for transfection protocol for some plasmids in materials and method section. How many micrograms of each plasmid? How many cells where plated and how many hours before the transfection?

F) This is probably some missing knowledge from me.

I have difficulty to understand how the authors can ascertain that BAP1 deubiquitinates KEAP1 by analysing Myc detection by Western blot. I see the differences, but I don’t know how this assay/analysis allow to assure the direct deubiquitnation of KEAP1 by BAP1. But as I said, probably I don’t know too much about this. Please highlight me about this issue.

G) Although it is quite normal and known for some researchers, it can be hard to understand by clinicians or researchers from other fields what CHX does and why it is used. Please just add between parentheses (inhibits protein synthesis), or something similar.

H) Why there are so many bands in some lanes of figure 2B, right? It is a question of unspecific binding to agarose beads or alternative splicing proteins due to the constructed used to transfect in those cells?

I) In line with H. In figure 3B, to disclose the direct correlation of BAP1 deubiquination of KEAP1 in lung adenocarcinoma cells, do the authors can IP KEAP1 and do a immunoblot against total ubiquitin to evaluate ubiquination levels of KEAP1?

J) IN section 3.3. the authors make a big jump from the molecular pathway BAP1-KEAP1-NRF2 target genes to biological/functional assays – migration, invasion and resistance to apoptosis induced by cisplatin. Please add an additional paragraph to make a transition from these points, how are they related and why do authors evaluate those cancer cell features.

K) In section 3.4, the authors argue that their results show that KRAS activation is responsible for BAP1 and KEAP1 downregulation. I have two questions:

K1) Have the authors measured (directly or indirectly) KRAS activation in lung tissues at 1, 8, 24 and 40 weeks, the same timepoints they evaluated mRNA expression for Bap1, Keap1, Hmox1 and Nrf2? It would be relevant do show in order to state that there a direct effect of KRAS on Bap1 and Keap1.

K2) Perhaps this is a very naïve question, but other might have the same question. If the animals possess the KRAS G12D/+ genotype, does it mean that KRAS might be always (hyper)activated in animals, independently of the age of the animals? If so, why do the authors see a time dependent reduction in Bap1/Keap1 mRNA expression? It is because there is increased overactivation of KRAS? Or is it a time dependent effect related with tumor progression? If it is the last case, (once again), can we assure that is KRAS activation a driver for Bap1 and Keap1 downregulation?

L1) Please provide additional immunohistochemical images, some of them with higher magnification, where it is possible to clearly see the staining patterns inside the cells (nuclear, cytoplasmatic or membrane staining).

L2) Is it normal that the non-tumoral (called by many as “normal”) adjacent tissue has much less BAP1 and KEAP1 expression than in WT normal lung tissue? It does not seem to happen for 8-OHdG. What happens in G12D lungs far away from the tumors?

M) Please provide additional information about all the selected parameters chosen to plot the graphs (Cutoffs, matched or non-matched data from TCGA and GTEx?, which differential statistical methods were used to calculate p?) represented in figures 5c-d.

I am sorry for the long report. In my opinion the manuscript and the work is very good, so I am trying to help (as much as I can and I know) the authors to improve to the maximum the publication!

Great work!

Author Response

Reviewer 2

The Keap1/NRF2 pathway has been explored in several types of human cancer in the recent years. The authors found a novel regulatory protein for this pathway.

In general, the manuscript is well written, experiments are well designed to answer all the scientific questions raised in the study and results are well interpreted. It is a great work. I have minor concerns that need to be addressed by the authors to avoid some confusions, add extra information to other researchers in the field can replicate the experiments if they need and to make data clear to the readers. Please consider my comments as positive suggestions to improve the quality of the manuscript (scientifically, but also to help who will read the manuscript when published).

  1. A) In the materials and methods section, the authors should include the manufacturer’s references for the purchased reagents, whenever possible. Sometimes there are similar products with slight formulation differences; other example is the fact that there are several antibodies for the same protein from the same company; and so on. So please, whenever possible, please add the product reference.

Response

I would like thank this reviewer for critical and favorable comments. Complying with this reviewer’s comments, we have amended the sentences in the manuscript to reflect the comments and changes were highlighted in red color in the revised manuscript.

  1. B) The authors describe the antibodies used, but they should mention the dilution made. Please add this extra information!

Response

We had added the catalogue numbers of individual antibodies and described the dilution ratio in the materials and methods or in the figure legends.

  1. C) In the WB method description, I am not sure about these points.

C1) Usually the incubation with secondary antibody is made at room temperature. Do the authors incubate for 1 h at 4ºC as described?

C2) Do the authors dilute primary antibodies only in 1x PBS? Or do they dilute in 5% non-fat milk (in 1x PBS)?

C3) Which ECL system do authors use? Please add the company and reference of the product.

C4) Please add antibody dilutions (primary and secondary antibodies).

Response

C1) We conducted hybridization of the membrane with secondary antibodies for 1 h at 4ºC.

C2) We conducted hybridization of the membrane with primary antibodies usually overnight at 4ºC in 1x PBS.

C3) We generated an in-house ECL solution according to our lab protocol. This ECL solution works pretty well.

C4) We have added antibody dilution ratios in the manuscript.

  1. D) There is a missing information in the material and methods section for some plasmids used. For instance, there is no mention to the HA-NRF2 construct (figure 1C). Also for some others.

Response

We have acquired most of cDNAs by RT-PCR amplification from cell lines, verified the sequence, and utilized them for subcloning plasmids.

  1. E) Please add additional information for transfection protocol for some plasmids in materials and method section. How many micrograms of each plasmid? How many cells where plated and how many hours before the transfection?

Response

We have revised the Materials and Methods or the Figure legends to reflect the comments of this reviewer. Please refer to the revised manuscript.

  1. F) This is probably some missing knowledge from me.

I have difficulty to understand how the authors can ascertain that BAP1 deubiquitinates KEAP1 by analysing Myc detection by Western blot. I see the differences, but I don’t know how this assay/analysis allow to assure the direct deubiquitnation of KEAP1 by BAP1. But as I said, probably I don’t know too much about this. Please highlight me about this issue.

Response
The activity of poly-ubiquitination can be assumed by the intensity of Myc-ubiquitin conjugated to the immunoprecipitated protein in the samples.

  1. G) Although it is quite normal and known for some researchers, it can be hard to understand by clinicians or researchers from other fields what CHX does and why it is used. Please just add between parentheses (inhibits protein synthesis), or something similar.

Response

To reflect the opinion of this reviewer, we have added the sentence that CHX was used to inhibit de novo protein synthesis.

  1. H) Why there are so many bands in some lanes of figure 2B, right? It is a question of unspecific binding to agarose beads or alternative splicing proteins due to the constructed used to transfect in those cells?

Response

We still do not know why it happens, but selected DUBs seem to undergo post-translational modifications and/or proteolytic cleavage.

  1. I) In line with H. In figure 3B, to disclose the direct correlation of BAP1 deubiquination of KEAP1 in lung adenocarcinoma cells, do the authors can IP KEAP1 and do a immunoblot against total ubiquitin to evaluate ubiquination levels of KEAP1?

Response

We tried these experiments, but the affinity of antibodies we purchased and used in the manuscript was not sensitive enough to conduct endogenous immunoprecipitation followed by Western blot with ubiquitin antibody.

  1. J) IN section 3.3. the authors make a big jump from the molecular pathway BAP1-KEAP1-NRF2 target genes to biological/functional assays – migration, invasion and resistance to apoptosis induced by cisplatin. Please add an additional paragraph to make a transition from these points, how are they related and why do authors evaluate those cancer cell features.

Response

We conducted these experiments because the inhibition of NRF2 activity could suppress tumorigenic activity and, thus, sensitize A549 cells to cisplatin-induced apoptosis We have indicated this issue in the manuscript..

  1. K) In section 3.4, the authors argue that their results show that KRAS activation is responsible for BAP1 and KEAP1 downregulation. I have two questions:

K1) Have the authors measured (directly or indirectly) KRAS activation in lung tissues at 1, 8, 24 and 40 weeks, the same timepoints they evaluated mRNA expression for Bap1, Keap1, Hmox1 and Nrf2? It would be relevant do show in order to state that there a direct effect of KRAS on Bap1 and Keap1.

Response

K1) Because the activation of KRAS occurs by genetic mutation in Kras, there is no way to directly measure the activation of KRAS in vivo and we assumed that KRAS is constitutively activated.

K2) Perhaps this is a very naïve question, but other might have the same question. If the animals possess the KRAS G12D/+ genotype, does it mean that KRAS might be always (hyper)activated in animals, independently of the age of the animals? If so, why do the authors see a time dependent reduction in Bap1/Keap1 mRNA expression? It is because there is increased overactivation of KRAS? Or is it a time dependent effect related with tumor progression? If it is the last case, (once again), can we assure that is KRAS activation a driver for Bap1 and Keap1 downregulation?

Response

K2) Yes, KRAS is constitutively activated in vivo due to substitution mutation in KRAS. In addition, there exist a paper claiming that KRAS activation can lead to NRF2 activation although the underlying mechanisms were uncertain. We believe that the reduction of Bap/Keap1 by accumulative KRAS activation is attributable to how NRF2 activation occurs in lung tumors.

L1) Please provide additional immuno histochemical images, some of them with higher magnification, where it is possible to clearly see the staining patterns inside the cells (nuclear, cytoplasmatic or membrane staining).

Response

L1) Complying with the comments of this reviewer, we have changed the immunohistochemistry images in the revised manuscript.

L2) Is it normal that the non-tumoral (called by many as “normal”) adjacent tissue has much less BAP1 and KEAP1 expression than in WT normal lung tissue? It does not seem to happen for 8-OHdG. What happens in G12D lungs far away from the tumors?

Response

L2) We still don’t know how lung tumors arise in selected regions in the lung of mice. but we see that adjacent tissues around lung tumors had a less BAP1 and KEAP1 expression.

  1. M) Please provide additional information about all the selected parameters chosen to plot the graphs (Cutoffs, matched or non-matched data from TCGA and GTEx?, which differential statistical methods were used to calculate p?) represented in figures 5c-d.

Response)

To clarify this issue, we have revised the Materials and Methods and in Figure legends.

I am sorry for the long report. In my opinion the manuscript and the work is very good, so I am trying to help (as much as I can and I know) the authors to improve to the maximum the publication!

Great work!